# Artificial fingerprints for cross-comparison of forensic DNA and protein recovery methods

**Danielle S. LeSassier**[1], **Kathleen Q. Schulte**[1], **Tara E. Manley**[1], **Alan R. Smith**[1], **Megan L. Powals**[1], **Nicolette C. Albright**[1], **Benjamin C. Ludolph**[1], **Katharina L. Weber**[1], **August E. Woerner**[2,3], **Myles W. Gardner**[1], **F. Curtis Hewitt**[1]*

**1** Signature Science, LLC, Austin, Texas, United States of America, **2** Center for Human Identification, University of North Texas Health Science Center, Fort Worth, Texas, United States of America, **3** Graduate School of Biomedical Sciences, University of North Texas Health Science Center, Fort Worth, Texas, United States of America

* chewitt@signaturescience.com

**Data Availability Statement:** Data underlying this study have been deposited in the PRoteomics IDEntifications (PRIDE) database under dataset

## Abstract

Quantitative genomic and proteomic evaluation of human latent fingerprint depositions represents a challenge within the forensic field, due to the high variability in the amount of DNA and protein initially deposited. To better assess recovery techniques for touch depositions, we present a method to produce simple and customizable artificial fingerprints. These artificial fingerprint samples include the primary components of a typical latent fingerprint, specifically sebaceous fluid, eccrine perspiration, extracellular DNA, and proteinaceous epidermal skin material (i.e., shed skin cells). A commercially available emulsion of sebaceous and eccrine perspiration material provides a chemically-relevant suspension solution for fingerprint deposition, simplifying artificial fingerprint production. Extracted human genomic DNA is added to accurately mimic the extracellular DNA content of a typical latent print and comparable DNA yields are recovered from the artificial prints relative to human prints across surface types. Capitalizing on recent advancements in the use of protein sequence identification for human forensic analysis, these samples also contain a representative quantity of protein, originating from epidermal skin cells collected from the fingers and palms of volunteers. Proteomic sequencing by liquid chromatography-tandem mass spectrometry (LC-MS/MS) analysis indicates a high level of protein overlap between artificial and latent prints. Data are available via ProteomeXchange with identifier PXD015445. By including known quantities of DNA and protein into each artificial print, this method enables total DNA and protein recovery to be quantitatively assessed across different sample collection and extraction methods to better evaluate extraction efficiency. Collectively, these artificial fingerprint samples are simple to make, highly versatile and customizable, and accurately represent the biochemical composition and biological signatures of human fingerprints.

identifier PXD015445. All other relevant data are within the paper.

**Funding:** This research is based upon work supported in part by the Office of the Director of National Intelligence (ODNI), Intelligence Advanced Research Projects Activity (IARPA) (https://www.iarpa.gov), via contract number 2018-18041000003. The views and conclusions contained herein are those of the authors and should not be interpreted as necessarily representing the official policies, either expressed or implied, of ODNI, IARPA, or the U.S. Government. The U.S. Government is authorized to reproduce and distribute reprints for governmental purposes notwithstanding any copyright annotation therein. The funders reviewed and approved the manuscript for publication but had no role in study design, data collection and analysis, or preparation of the manuscript. This research was supported in part by internal funding from Signature Science, LLC. Signature Science, LLC provided support in the form of salaries for authors DSL, KQS, TEM, ARS, MLP, NCA, BCL, KLW, MWG, and FCH but did not have any additional role in the study design, data collection and analysis, decision to publish, or preparation of the manuscript. The specific roles of these authors are articulated in the 'author contributions' section.

**Competing interests:** The authors DSL, KQS, TEM, ARS, MLP, NCA, BCL, KLW, MWG, and FCH are employed by Signature Science, LLC. This does not alter our adherence to PLOS ONE policies on sharing data and materials. The authors declare no other relevant affiliations or financial involvement with a financial interest in or financial conflict with the subject matter or materials discussed in the manuscript apart from those disclosed.

# Introduction

Inter-replicate and inter-subject variability between human donor touch depositions poses a significant hurdle to the evaluation of emerging forensic analysis kits [1]. As no two latent fingerprint samples are the same and can vary depending on intrinsic individual characteristics, such as age, gender, and health, and deposition conditions, quantitatively defining the extraction efficiency of DNA or protein from a sample is problematic [2]. This has led to numerous comparisons in which sample extraction methods are evaluated on the frequency of successful analysis (e.g., generation of a complete DNA profile) rather than a quantitative evaluation of recovery [3].

Sampling DNA from touch samples presents a significant technical challenge and represents a growing number of forensic casework samples analyzed annually in the U.S. [4,5]. Additionally, the incorporation of protein for forensic analysis is a relatively new development. This method permits the identification of nonsynonymous mutations by peptide analysis in samples with little to no DNA (e.g., hair shafts) or degraded DNA [6–9]. The production of artificial fingerprints that accurately mimic the biochemical composition and biological signatures of human latent prints is an important step in defining the quantity and quality of forensic biomarkers extracted from touch samples.

In addition to the DNA and protein components, the chemical matrix of fingerprint depositions needs to be considered. While various biomolecular components of sebaceous and eccrine fluids are applicable for forensic analysis (e.g., determination of fingerprint age), these signatures have not been established for human identification [9]. Previously, an artificial sebaceous and eccrine fluid has been created to simulate the composition of a human touch sample [10]. This study established the emulsification process needed to combine sebaceous and eccrine fluids and enabled the incorporation of additional materials (e.g., explosive residue) into the artificial prints. These methods are compatible with latent fingerprint deposition tools capable of incorporating a ridge pattern for visualization and analysis [10–12]. While highly customizable, such mixtures can be laborious to replicate, and a simplified solution was sought.

The high variability in human fingerprints highlights the need for a defined, artificial standard. Here, artificial fingerprints are generated that contain the primary components of human latent fingerprints: sebaceous fluid, eccrine perspiration, extracellular DNA, and shed skin cells [2,10,13,14]. The similarity of these artificial fingerprints to human fingerprint depositions is demonstrated in terms of the sebaceous matrix, DNA and protein recovery, and proteomic analysis across two forensically-relevant surface types, supporting their use as a surrogate for human touch depositions. These artificial fingerprints enable accurate quantitation of DNA and protein recovery for the analysis of forensically relevant DNA markers (e.g., short tandem repeats, STRs, and single nucleotide polymorphisms, SNPs) and protein markers (e.g., genetically variable peptides, GVPs) in a commercially available sebaceous matrix while providing the versatility and customization to incorporate other forensically-relative components.

# Methods

## Collection of human epidermal skin material and fingerprint samples

A commercial skin exfoliation product (Ped Egg® Easy Curve™) was used to collect epidermal skin material (ESM) from the hands and fingers of volunteer skin donors. Prior to use, each Ped Egg tool was decontaminated by soaking in RNase Away (Molecular BioProducts) for 10 minutes, followed by a rinse with deionized water and 70% isopropyl alcohol, and then allowed to dry. Donors washed and dried hands immediately prior to donating skin. Donors then

exfoliated their skin by gently rubbing the Ped Egg exfoliating grate across the palm and fingers of each hand for 90 seconds, holding the grate facing upward and hand facing downward to collect the skin particles into the collection chamber below the exfoliating grate. Skin material was then collected under sterile conditions from each Ped Egg by removing the grate and using a decontaminated eyebrow brush (Maybelline) to gently brush the skin particles from the grate-holding collection chamber onto weigh paper and into a microcentrifuge tube.

Donors also provided latent and loaded fingerprint deposition samples on glass and chrome metal surfaces. Loaded fingerprints were acquired by having the donor rub their fingers across their neck and upper back areas prior to touching the surface to 'load' the deposition with additional sebaceous, proteinaceous, and DNA material. The donor then touched the surface for 10 seconds to deposit a loaded fingerprint. Latent fingerprints were acquired by having the donor wash their hands and directly touch the surface for 10 seconds without prior 'loading' of the sample.

Twenty-five adult (over 18 years old) male and female donors of northern European ancestry provided ESM samples for proteomic analysis by Ped Egg collection, as described above, and stored at -80 ˚C. All human subject material was collected using protocols and informed consents that were approved by the Institutional Review Boards at the University of North Texas Health Science Center (IRB #00642). Consent was obtained orally and in writing.

## Artificial fingerprint production

Collected donor ESM was lightly homogenized using a cell dissociation sieve (Sigma-Aldrich), with a size 60 mesh screen on top and a size 40 mesh screen on bottom, using a pestle to grind the material through the sieve onto weigh paper for collection. The homogenized ESM was collected, stored at -80 ˚C, and used for all subsequent artificial fingerprint preparations.

Homogenized donor ESM material was weighed (~1–2 mg) and resuspended in stabilized artificial eccrine perspiration (Pickering Labs, P/N 1700–0024) to a stock concentration of 25 µg/µL in a 2-mL LoBind protein microcentrifuge tube (Eppendorf). DNA stock solutions were commercially obtained at 100 ng/µL (Quantifiler Trio DNA Standard, Applied Biosystems). To prepare the final artificial fingerprint solutions, appropriate volumes of the 25 µg/µL ESM stock solution, 100 ng/µL DNA stock solution, and artificial eccrine perspiration-sebum emulsion (Pickering Lab, P/N 1700–0547), which is 5% sebum per the manufacturer, were diluted with stabilized artificial eccrine perspiration to final concentrations of 2.5 µg/µL of ESM, 2.0 ng/µL of DNA, and 2.5% sebum. The total volume of the artificial fingerprint solution was dependent on application and number of artificial fingerprints needed. Typically, 100 µL of artificial fingerprint solution was prepared by adding 50.0 µL of artificial eccrine perspiration-sebum emulsion (5% sebum), 10.0 µL of 25 µg/µL ESM stock, 2.0 µL of the 100 ng/µL Quant Trio DNA, and 38.0 µL of stabilized artificial eccrine perspiration. The artificial fingerprint solution was made fresh each time and used immediately for surface depositions.

Glass slides and chrome metal plates were decontaminated by soaking in RNAse Away (Molecular BioProducts) for 10 minutes, followed by deionized water and 70% isopropanol rinses, and then allowed to dry overnight prior to use. The artificial fingerprint solution was mixed well prior to each deposition by low speed vortexing or pipetting up and down. To deposit an artificial fingerprint, 5.0 µL of the artificial fingerprint solution was spotted onto a decontaminated surface of interest and allowed to dry overnight before processing. Following artificial print deposition, surfaces were kept in decontaminated containers at room temperature prior to collection for up to two weeks. For reference, a 5.0 µL spot deposits 12.5 µg of ESM and 10.0 ng of DNA.

## Surface collection and sample preparation for DNA analysis

For DNA collection, artificial, loaded, and latent fingerprint samples were collected from both surface types by rubbing a dry flocked swab (Puritan Hydraflock) in a serpentine motion vertically and horizontally across the deposition area. The swab tip was released into a Lyse & Spin basket (Qiagen) and DNA was extracted using a Qiagen DNA Investigator standard swab protocol with a 40 µL elution. DNA was quantified (Applied Biosystems Quantifiler™ Trio) and STRs were amplified (Applied Biosystems Veriti™) using Applied Biosystems GlobalFiler™ 29-cycle chemistry with a target input of 0.5 ng of DNA. STR fragments were detected by capillary electrophoresis (Applied Biosystems 3500xL genetic analyzer) with a 24-second injection time, and STR profiles were interpreted (GeneMapper™ v1.5) using an analytical threshold of 125 relative fluorescence units (RFU) and stochastic threshold of 525 RFU.

## SDS-PAGE analysis

Samples for SDS-PAGE gel analysis were prepared in a combination of NuPAGE LDS sample buffer (Invitrogen) and NuPAGE sample reducing agent (Invitrogen) per the manufacturer's protocol, boiled for 5 minutes, and loaded onto a NuPAGE 4–12% Bis-Tris protein gel (Invitrogen). Gels were run in NuPAGE MOPS SDS running buffer (Invitrogen) with NuPAGE antioxidant (Invitrogen) added to the inner chamber buffer at 200 V for 50 minutes. Gels were stained using SimplyBlue SafeStain (Invitrogen) and destained per the manufacturer's protocol. Novex sharp pre-stained protein standard (Invitrogen) was used.

## Surface collection and sample preparation for LC-MS/MS analysis

Artificial, latent, and loaded fingerprint samples were collected from glass and chrome metal surfaces. For both surfaces, 100 µL of 0.1% (w/v) RapiGest (Waters) in 50 mM ammonium bicarbonate was added directly to the fingerprint sample and allowed to sit on the surface for 5 minutes. For collection from glass, a shortened cell lifter (Fisher Scientific) was used to scrape the remaining liquid into a 50-mL conical tube, centrifuged for 3 minutes at $1,000 \times g$, and transferred to a LoBind protein microcentrifuge tube. For collection from metal, the remaining liquid on the metal surface was pipetted up-and-down over the fingerprint area, and the entire volume was collected and transferred into a LoBind protein microcentrifuge tube. Following surface-specific collection, all samples were brought up to 100 µL volume with 0.1% (w/v) RapiGest before proceeding.

For LC-MS/MS analysis, samples were lysed by water bath sonication for 1 minute, boiled for 5 minutes, cooled on ice, and briefly vortexed. Protein concentrations were measured by Qubit protein assay on a Qubit fluorometer (Thermo Fisher) per manufacturer's protocol. To the sample, 5.0 µL of 5.0 mg/mL dithiothreitol (DTT) in 50 mM ammonium bicarbonate was added and the solution was incubated at 65 ˚C for 15 minutes. The reduced disulfide bonds were alkylated by adding 5.0 µL of 15.0 mg/mL iodoacetamide (IAA) in 50 mM ammonium bicarbonate and allowing the reaction to proceed for 30 minutes in the dark at room temperature. Trypsin (Promega) was reconstituted in 50 mM ammonium bicarbonate, added at a 1:30 w/w ratio to the protein sample based on previously measured Qubit values, and incubated for 16 h at 37 ˚C. Following trypsin enzymatic digestion, RapiGest was precipitated with 0.5% trifluoracetic acid and the supernatant was passed through a 0.2 µm centrifugal filter (Thermo Fisher). The flow-through was dried in a Vacufuge (Eppendorf) and resuspended in 50 mM acetic acid prior to analysis by mass spectrometry.

## Mass spectrometry (LC-MS/MS and GC-MS)

The fingerprint peptide samples were analyzed on a Thermo Scientific Q Exactive Plus high-resolution, accurate-mass (HRAM) mass spectrometer coupled to a Thermo Scientific Ultimate 3000 nano LC system. Samples were injected onto an Acclaim PepMap 100 loading column (75 μm × 2 cm, Thermo Scientific), then transferred to a PepMap RSLC C18 column (75 μm × 25 cm, Thermo Scientific) for analysis. The column temperature was held at 45 ˚C. Mobile phase A was comprised of 98: 2 water: acetonitrile (v/v) with 0.1% formic acid, and mobile phase B was comprised of 98: 2 acetonitrile: water (v/v) with 0.1% formic acid. The mobile phases were maintained at a flow rate of 300 nL/min. The solvent gradient started at 98% A, held for 5 minutes, and was then linearly ramped over 200 minutes to 80% A, to 68% A over 50 minutes, to 5% A in 40 minutes, and finally held at 5% A for 5 minutes. The column was then re-equilibrated at 98% A for 20 minutes prior to starting the next run. The LC eluent was ionized using a nanospray (NSI) source in positive ion mode.

The mass spectrometer was operated in data dependent $MS^2$ mode. Full scan mass spectra were acquired from $m/z$ 375 to 1575 at a resolution of 70,000. The fifteen (15) most abundant precursor ions in each full MS1 spectrum (intensity threshold of 50,000) were selected for fragmentation using an isolation width of 1.6 $m/z$ and a normalized collision energy of 17,500. Tandem mass spectra were acquired at a resolution of 17,500.

## Proteomic data analysis

After analysis by nanoLC-HRAM-MS/MS, data from the samples were further analyzed using Proteome Discoverer 2.1 SP1 (Thermo Fisher). Data files were processed using SequestHT and Percolator analytical tools against a custom generated FASTA file based on the UniProt human database, augmented with specific peptides and proteins that were targets of this study. The precursor mass tolerance was set to 20 ppm and the fragment ion mass tolerance to 0.8 Da. The enzyme was set to trypsin with a maximum of two missed cleavages. Cysteines were fixed with carbamidomethyl modifications and the only allowable variable modification was methionine oxidation. Peptide spectral matches (PSMs), peptide identifications, and protein identifications were all validated using Percolator at a false discovery rate of 0.05. Outputs of Proteome Discoverer were further processed and summarized in R (v3.5.1). The mass spectrometry proteomics data have been deposited to the ProteomeXchange Consortium [15] via the PRIDE [16] partner repository with the dataset identifier PXD015445 and 10.6019/PXD015445.

## Results

### Similarity of commercial sebum/perspiration mixture

While the components in human eccrine (i.e., sweat) and sebaceous fluids present in touch samples have been extensively characterized [2,10,17,18], there is no current commercial source that has been evaluated for mimicking the components identified in such previous studies. Emulsification of the hydrophobic and hydrophilic components of such a mixture is challenging, and the stability of artificial sebaceous solutions has not been fully established. To overcome these issues, a commercially available eccrine/sebaceous fluid mixture was identified from Pickering Labs that is stable under standard storage conditions, thus simplifying the artificial fingerprint production process.

Per the manufacturer, the sebum percentage in the commercial mixture is 5%. To enable customization (e.g., addition of DNA) in the artificial fingerprint solution while still

maintaining a known sebum concentration, the sebum was diluted to 2.5% for all artificial fingerprints using stabilized eccrine perspiration from Pickering Labs as the diluent.

To ensure that the premade solution has similar biochemical properties to human fluids, the chemical composition relative to that of human fingerprints was compared using relevant literature (Table 1). The artificial solutions contain many of the published components present in human fingerprints, including multiple saturated and unsaturated fatty acids, various amino acids, inorganic salts and minerals, and other signature compounds, such as squalene and cholesterol. While not comprehensive, the relatively high overlap of major chemical components suggests the artificial sebaceous/eccrine mixture is an acceptable matrix surrogate for artificial fingerprints.

## Comparison of DNA content and quality

Extracellular DNA forms the majority of DNA present in a latent fingerprint sample [13,14,19,20]. To accurately represent this extracellular DNA, an extracted human genomic DNA standard was used. This allowed the addition of known DNA quantities in each artificial fingerprint. To approximate latent and loaded fingerprints, 5 or 10 ng of total DNA was spiked in to the artificial solutions, respectively. Latent, loaded, and artificial prints (containing either 5 or 10 ng DNA) were deposited onto glass or chrome metal surfaces, two non-porous surfaces of forensic value, collected, and extracted for DNA quantitation.

As shown in Fig 1A, DNA from artificial fingerprints was successfully recovered from glass and metal surfaces in ranges that generally matched the DNA yield ranges from the human prints. DNA recovery from artificial fingerprints on glass showed better correlation to the corresponding human prints than from metal, where the values were slightly elevated. Additionally, the artificial fingerprints showed less overall variability in terms of total DNA yield compared to their human counterparts (Glass—latent 0.96 ± 0.65 ng, loaded 2.46 ± 0.74 ng, AF (10) 1.84 ± 0.43 ng, AF (5) 1.05 ± 0.06 ng; Metal—latent 0.65 ± 0.55 ng, loaded 2.13 ± 1.47 ng, AF (10) 4.88 ± 1.47 ng, AF (5) 2.30 ± 0.32 ng).

To establish the relative DNA quality in human and artificial prints, DNA from each type of print was compared by a degradation index (DI) using the Quantifiler Trio kit. The DI is a ratio that compares the quantification value of a small and large amplicon within the same sample. DI values greater than 1 indicate DNA degradation [21]. DI values for both human and artificial fingerprint samples are comparable across glass and metal surfaces, and no major DNA degradation is apparent in the samples (Fig 1B). This suggests that the use of extracted genomic DNA in artificial fingerprints does not result in significant bias in the quality of DNA relative to DNA from human deposits.

## Processing collected ESM for artificial prints and evaluation of protein recovery

The volunteer-collected epidermal skin material (ESM) presented a challenge in the irregular nature of the skin particles, making sample manipulation and pipetting difficult (Fig 2A, left). To address this, donor ESM was passed through a fine mesh sieve to filter out large particulate and lightly homogenize the material. This created finer skin particles (Fig 2A, right), increasing the ESM manipulability for artificial fingerprint generation. While this step greatly improved pipetting the artificial fingerprints, it did not completely alleviate issues associated with larger ESM particulates causing clogging or uneven deposition. Light microscopy was used to visualize the skin particle sizes in latent, loaded, and artificial fingerprints (Fig 2B). Similarities in particulate size were observed between loaded and artificial prints, as well as between artificial prints and the larger ESM fragments present in latent depositions.

**Table 1. Chemical components in human and artificial fingerprints.**

| Human Fingerprints* | Artificial Fingerprints† |
|---|---|
| **Fatty Acids** | |
| Arachidic acid (eicosanoic acid) [10,18] | |
| Caprylic acid (octanoic acid) [10,17,18] | |
| | Coconut oil (caprylic acid, lauric acid, myristic acid, oleic acid, palmitic acid)‡ |
| Linoleic acid (9,12-octadecadienoic acid) [10,18] | Linoleic acid, pure |
| Myristic acid (tetradecanoic acid) [10,17,18] | |
| Oleic acid (octadecenoic acid) [10,17,18] | Oleic acid, pure |
| Palmitic acid (hexadecanoic acid) [2,10,17,18] | Palmitic acid (cetylic acid) |
| Palmitoleic acid (9-hexadecenoic acid) [10,17] | |
| Stearic acid (octadecanoic acid) [2,10,17,18] | Stearic acid, pure |
| | Virgin olive oil (linoleic acid, oleic acid, palmitic acid, palmitoleic acid, stearic acid)‡ |
| **Amino Acids** | |
| Alanine [18] | Alanine |
| | Arginine |
| Asparagine [18] | Asparagine |
| Aspartic acid [18] | Aspartic acid |
| | Citrulline |
| Cysteine [18] | |
| Glycine [10,18] | Glycine |
| Glutamic acid [18] | Glutamic acid |
| | Histidine |
| Isoleucine [18] | Isoleucine |
| Leucine [18] | Leucine |
| Lysine [18] | Lysine |
| | Methionine |
| Ornithine [18] | Ornithine |
| Phenylalanine [18] | Phenylalanine |
| Serine [10,18] | Serine |
| | Threonine |
| Tyrosine [18] | Tyrosine |
| Valine [10,18] | Valine |
| **Inorganic Salts and Minerals** | |
| Calcium [2,10] | Calcium |
| Chloride [2,10] | Chloride |
| | Copper |
| Iron [10] | Iron |
| Magnesium [2,10] | Magnesium |
| | Nitrates |
| Sodium [2,10] | Sodium |
| | Zinc |
| **Other Components** | |
| Cholesterol [2,10,17] | Cholesterol |

*(Continued)*

**Table 1.** (Continued)

| Human Fingerprints* | Artificial Fingerprints† |
|---|---|
| Lactic acid [2] | Lactic acid |
| Large hydrocarbons [17] | Large hydrocarbons (Paraffin waxes) |
| Squalene [2,10,17,18] | Squalene (2,6,10,15,19,23-hexamethyltetracosa-2,6,10,14,18,22-hexaene) |
| Urea [2,10] | Urea |
| Uric acid [2] | Uric acid |

A list of select compounds present in human and artificial fingerprints. Alternative compound names are given in parentheses where applicable.

* Select references for human fingerprint compounds are listed.

† Compounds for artificial fingerprints are from the Pickering Laboratories product page (https://www.pickeringtestsolutions.com/AP-eccrine/) and Safety Data Sheets for P/N 1700–0547 and 1700–0024.

‡ Top five fatty acids as listed in the U.S. Department of Agriculture FoodData Central (https://fdc.nal.usda.gov/). Olive oil, FDC ID 171413; Coconut oil, FDC ID 171412.

To ensure the amount of ESM loaded into the artificial fingerprints resulted in protein levels and profiles comparable to a typical latent fingerprint deposition, a range-finding experiment was performed. Homogenized ESM was measured, diluted in the sebum-eccrine mixture, and deposited at the indicated amounts (2.5, 5.0, 7.5, 10.0, or 12.5 μg ESM) on a glass slide for collection. Artificial and latent fingerprint samples were collected and run on an SDS-PAGE gel for analysis (Fig 2C). While some variability was observed between replicates, possibly due to larger particulate deposition not completely removed during the homogenization step, the 12.5 μg deposition appeared to most closely match the protein levels in a latent fingerprint. Further, the prominent bands in the latent prints are present in the artificial fingerprints (Fig 2C, arrowheads), suggesting that the protein profile of the collected ESM is not drastically different than that of a latent deposition. A comprehensive comparison of the protein profiles between artificial and latent prints is presented below.

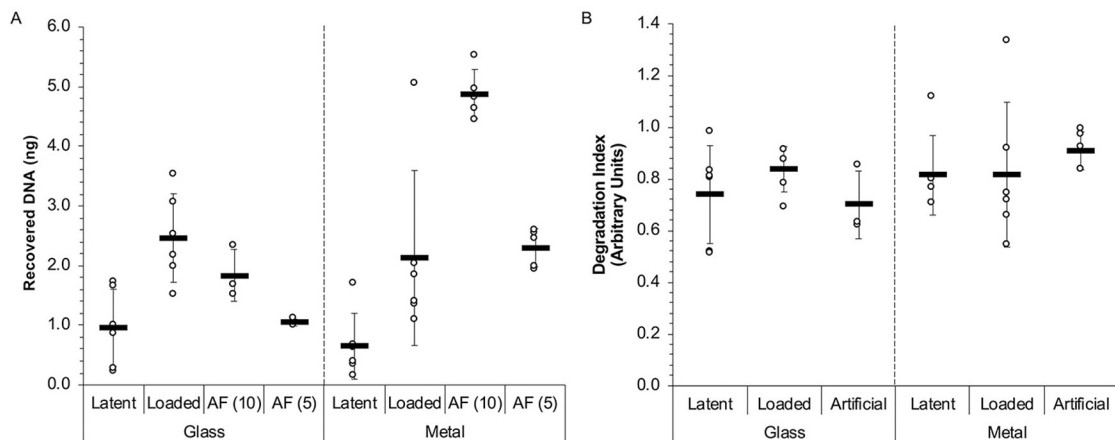

**Fig 1. Comparison of DNA yield and quality in latent and artificial fingerprints.** (A) Latent, loaded, and artificial fingerprints were deposited on two surfaces followed by DNA extraction to evaluate the total yield. (B) Comparison of DNA degradation index (DI) across fingerprint deposition on multiple surface types, where a DI ratio of greater than 1.0 indicates DNA degradation. AF (10), artificial fingerprints with 10 ng DNA; AF (5), artificial fingerprints with 5 ng DNA. Individual replicates are shown (circles) with the mean (bar) ± SD. For latent and loaded fingerprint samples, from both metal and glass, $n$ = 6. For both types of artificial fingerprints, $n$ = 3 for samples from glass and $n$ = 5 for samples from metal.

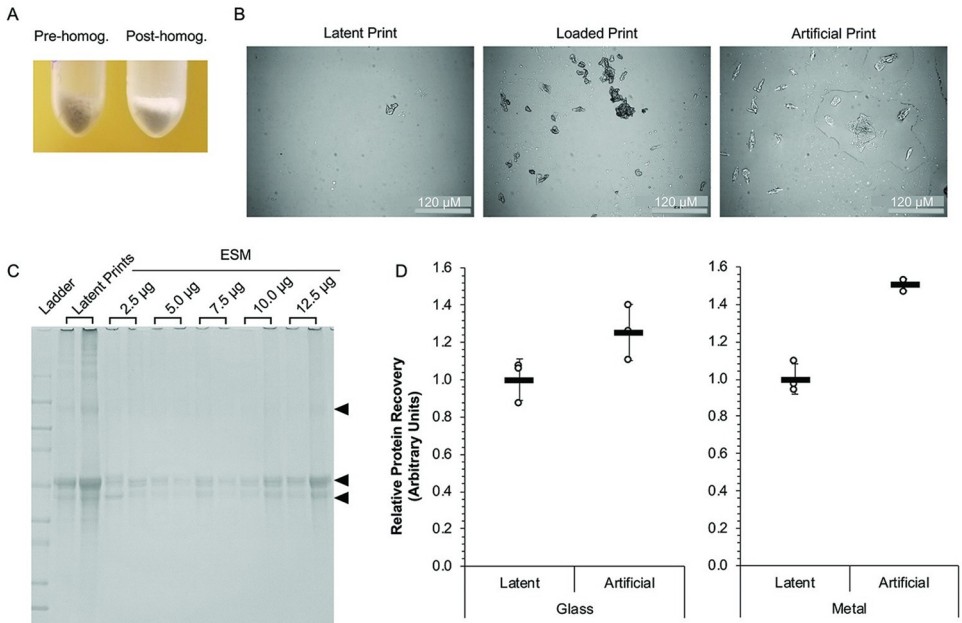

**Fig 2. Comparison of protein yield and quality between latent and artificial fingerprints.** (A) Comparison of ESM pre-homogenization (pre-homog., left) or following sieve-based homogenization (post-homog., right) shows reduction in the overall skin particle size. (B) Evaluation of ESM size in deposited latent (left), loaded (middle), or artificial (right) fingerprints on glass by light microscopy. (C) Representative SDS-PAGE results from an artificial fingerprint ESM range-finding experiment to determine the corresponding protein amount in typical latent fingerprints. Arrowheads indicate prominent bands found in both latent and artificial fingerprints. (D) Protein recovery measured by a Qubit fluorometric assay between latent and artificial fingerprint samples across two surface types. The amount of protein recovered was quantified and the relative amount normalized to the surface-specific latent print average. Individual replicates (*n* = 3) are shown (circles) with the mean (bar) ± SD.

To evaluate protein recovery from different surfaces, artificial and latent prints were deposited on glass or chrome metal and collected by a surface-appropriate method. Protein recovery was determined by Qubit analysis and the values normalized on the average protein recovered from the latent prints on a given surface (Fig 2D). Protein recovery was largely consistent between replicates for artificial fingerprints, typically more so than latent depositions, indicating a higher level of reproducibility. As observed with DNA, protein recovery from artificial fingerprints on glass had better correlation to their latent counterparts then recovery from metal. Generally, the amount of protein recovered from artificial prints and latent prints was similar across both surfaces, supporting a deposition amount of 12.5 µg ESM.

## Comparison of protein content

To ensure that ESM collected for artificial fingerprints accurately recapitulates the proteomic profile present in latent fingerprints, a series of artificial and latent fingerprints were deposited on and extracted from glass and chrome metal surfaces. Following proteolytic digestion and nanoLC-HRAM-MS/MS analysis, sample data was analyzed using Proteome Discoverer in conjunction with a protein reference database based on the UniProt human proteome [22]. Table 2 lists the top 10 protein identifications determined by total number of peptide spectral matches (PSMs) to each protein from representative artificial and latent fingerprint samples collected from glass. While some variability exists in PSM count, the protein identifications are almost entirely in agreement. Proteins that are not concordant between lists (e.g.,

**Table 2. Top protein identifications by peptide spectral matches in artificial and latent fingerprint samples.**

| | Artificial Fingerprints | | Latent Fingerprints | |
|---|---|---|---|---|
| Rank | Protein | # PSMs | Protein | # PSMs |
| 1 | Keratin, type I cytoskeletal 9 (KRT9) | 367 | Keratin, type I cytoskeletal 10 (KRT10) | 221 |
| 2 | Keratin, type II cytoskeletal 1 (KRT1) | 259 | Keratin, type II cytoskeletal 1 (KRT1) | 170 |
| 3 | Keratin, type I cytoskeletal 10 (KRT10) | 102 | Keratin, type II cytoskeletal 2 (KRT2) | 103 |
| 4 | Keratin, type II cytoskeletal 2 (KRT2) | 96 | Keratin, type I cytoskeletal 9 (KRT9) | 93 |
| 5 | Hornerin (HRNR) | 76 | Keratin, type I cytoskeletal 14 (KRT14) | 38 |
| 6 | Keratin, type I cytoskeletal 14 (KRT14) | 37 | Keratin, type II cytoskeletal 1b (KRT77) | 32 |
| 7 | Keratin, type II cytoskeletal 5 (KRT5) | 37 | Keratin, type I cytoskeletal 16 (KRT16) | 31 |
| 8 | Keratin, type II cytoskeletal 6B (KRT6B) | 28 | Keratin, type II cytoskeletal 5 (KRT5) | 30 |
| 9 | Keratin, type I cytoskeletal 16 (KRT16) | 27 | Keratin, type II cytoskeletal 6B (KRT6B) | 23 |
| 10 | Desmoplakin (DSP) | 23 | Hornerin (HRNR) | 22 |

Desmoplakin) fell just outside the top-10 protein identification list. The complete list of protein identifications can be found in the supporting information (S1 Table).

To achieve a more global comparison of proteomic data between samples, the overall proteomic composition of latent and artificial fingerprint samples deposited on glass and metal surfaces were compared. Fig 3A illustrates the total protein sequence coverage and total number of peptides identified for each of the 50 most abundant proteins identified across all four samples. The overall sequence coverage and peptide counts are consistent across latent and artificial samples, suggesting that the protein collection method utilized to generate the ESM for

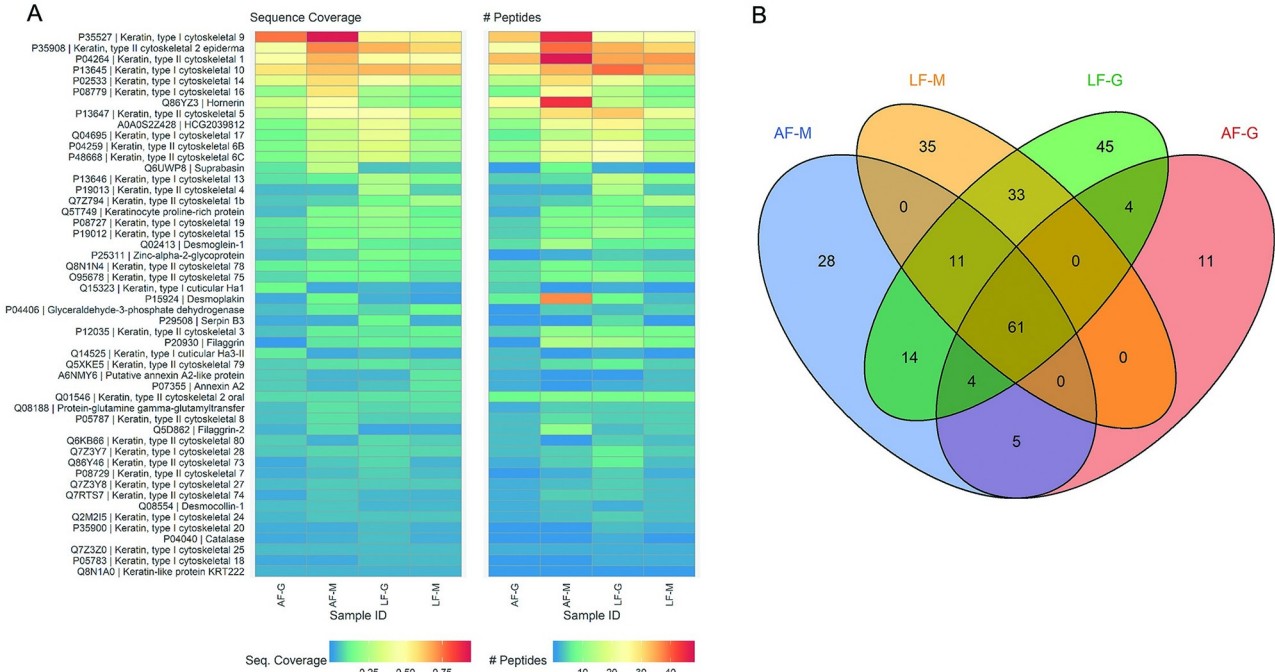

**Fig 3. Comparison of proteome composition between artificial and latent fingerprints.** (A) Protein sequence coverage (left) and number of peptides (right) detected for the fifty proteins with the highest mean sequence coverage detected in artificial or latent fingerprint samples. (B) Overlap of all proteins detected in artificial (AF) or latent (LF) fingerprint samples on metal (M) or glass (G) surfaces.

artificial fingerprints accurately recapitulates the protein content of human fingerprint deposits.

Fig 3B illustrates the overlap of identified proteins between artificial and latent fingerprint samples deposited on glass and metal surfaces. A total of 61 proteins were common to all four samples. These proteins are primarily keratins and other structural proteins common to corneocytes. A substantial number of protein IDs were unique to each individual sample (AF-M, 28; LF-M, 35; LF-G, 45; AF-G, 11). This is likely due to stochastic effects arising from low protein abundance in the original fingerprint sample, as well as data-dependent mass spectrometric analysis in which only the most abundant peptide precursor ions are selected for fragmentation and analysis [23]. Overall, the similarities in presence, sequence coverage, and number of peptides in the most abundant proteins between latent and artificial fingerprint samples supports concordance in the proteomic profiles.

## Discussion

We have successfully developed an artificial fingerprint solution that accurately mimics the composition of human latent prints using a commercial sebaceous/eccrine perspiration mixture, extracted genomic DNA, and donor-collected ESM (Fig 4). This method provides a mechanism for quantitatively assessing forensic collection and extraction tools by comparing the total yield of purified DNA and/or protein to a known starting amount. The artificial fingerprints described herein were designed to be straightforward to produce. Commercially available perspiration and sebaceous fluid provides a simple, shelf-stable solution that simulates the properties of human samples (Table 1). Use of extracted genomic DNA mirrors both the quantity and quality of DNA found in human samples (Fig 1) and can be sourced internally or obtained as a commercial standard. Protein may also be sourced internally, collected using a commercially available Ped Egg, or similar mechanical means to obtain epidermal skin material. This enables precise amounts of ESM to be weighed and added to the artificial

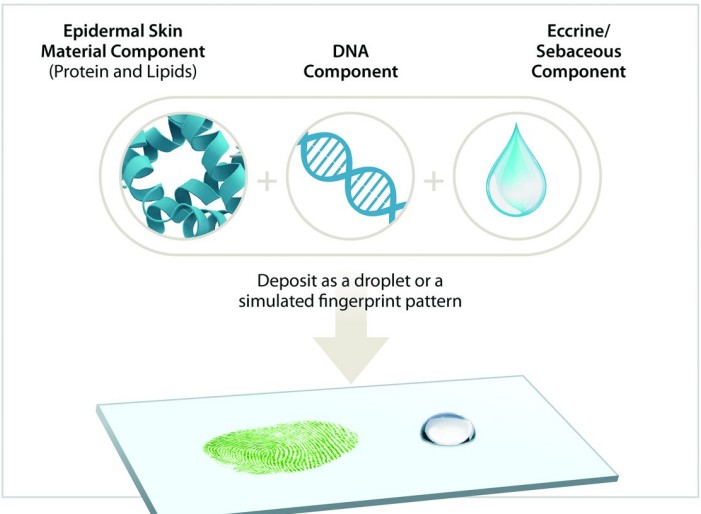

**Fig 4. Development of simple and customizable artificial fingerprints.** Artificial fingerprints developed herein incorporate both protein and DNA, making these versatile surrogates for method development of human forensic technologies focused on DNA (STR or SNP analysis) or protein (GVP analysis) markers, with the ability to be customized based on the research needs.

fingerprint (Fig 2). The ESM displays similar protein profiles to latent fingerprint depositions (Fig 3; Table 2), enabling accurate proteomic-based analysis.

The sebaceous matrix for the described artificial fingerprints was not a primary focus of this manuscript and the commercial products provide a consistent and relatively equivalent resuspension solution (Table 1). The literature indicates there is high variability in the lipid composition of fingerprint depositions [2,17,18,24], with recent work suggesting that individuals may fall into a "high" or "low" lipid donor category [17]. The use of commercial products was chosen to streamline the artificial print method, but if a more defined sebaceous mixture is desired the reader is referred to Sisco et al. [10].

The current study does not explore the stability of artificial fingerprints, in-solution or as a deposition, opting to make each sample fresh prior to collection. As such, it is currently unclear what potential impacts long-term storage may have on the integrity of the DNA or protein. Being able to stably store these artificial fingerprints without DNA or protein degradation is important for forensic application. Proteins are inherently more environmentally robust biomolecules than DNA [25,26], but how the artificial sebaceous/perspiration mixture affects either component under long-term storage conditions has not yet been defined. Stability studies of both in-solution and deposited artificial fingerprints are of future interest.

The intrinsic simplicity of artificial fingerprint assembly promotes reproducibility across production lots and enables tailored design, such as reducing DNA input to simulate fingerprints with little or no DNA or increasing DNA input to compensate for surface-dependent variability in recovery. While inherent sample variability is important in defining limitations of new methodologies, benchmarking quantitative metrics, such as extraction efficiency, requires a consistent and known input. Thus, the use of the described artificial fingerprints provides a complementary and enhancing method to current approaches. Further, these fingerprint samples can be customized by the addition of external components of forensic interest, such as explosive residue, microbial communities, or simulating human mixtures of both protein and DNA. This represents a strength of the artificial fingerprint method, as individual elements can be adjusted to accurately mimic a human fingerprint deposition for any circumstance of interest.

## Conclusions

The artificial fingerprint samples described here are simple to generate, customizable, compositionally mimic human touch depositions, and provide a way to accurately benchmark both commercial and novel collection and extraction tools across a variety of surfaces. This ensures that forensic casework laboratories are using the optimal method for generating human forensic markers from forensic samples.

## Supporting information

**S1 Table. Complete list of protein identifications for artificial and latent fingerprints.**
(XLSX)

## Acknowledgments

The authors would like to thank Dr. Andrew Reed, Maryam Baniasad, Dr. Liwen Zhang, and Dr. Michael Frietas at The Ohio State University for providing proteomic testing and feedback for artificial print samples. The authors also thank Dr. Bruce Budowle for technical feedback and assistance with IRB approvals.

## Author Contributions

**Conceptualization:** Danielle S. LeSassier, Kathleen Q. Schulte, Myles W. Gardner, F. Curtis Hewitt.

**Data curation:** Danielle S. LeSassier, Alan R. Smith, Myles W. Gardner, F. Curtis Hewitt.

**Formal analysis:** Danielle S. LeSassier, Kathleen Q. Schulte, Tara E. Manley, Nicolette C. Albright, Benjamin C. Ludolph, Myles W. Gardner, F. Curtis Hewitt.

**Funding acquisition:** Myles W. Gardner, F. Curtis Hewitt.

**Investigation:** Danielle S. LeSassier, Kathleen Q. Schulte, Tara E. Manley, Alan R. Smith, Megan L. Powals, Nicolette C. Albright, Benjamin C. Ludolph, Katharina L. Weber, Myles W. Gardner, F. Curtis Hewitt.

**Methodology:** Megan L. Powals, F. Curtis Hewitt.

**Project administration:** Kathleen Q. Schulte, August E. Woerner, Myles W. Gardner, F. Curtis Hewitt.

**Resources:** Kathleen Q. Schulte, Myles W. Gardner, F. Curtis Hewitt.

**Software:** Alan R. Smith, Myles W. Gardner, F. Curtis Hewitt.

**Supervision:** Danielle S. LeSassier, Kathleen Q. Schulte, Myles W. Gardner, F. Curtis Hewitt.

**Validation:** Danielle S. LeSassier, F. Curtis Hewitt.

**Visualization:** Danielle S. LeSassier, Myles W. Gardner, F. Curtis Hewitt.

**Writing – original draft:** Danielle S. LeSassier, Kathleen Q. Schulte, Alan R. Smith, Myles W. Gardner, F. Curtis Hewitt.

**Writing – review & editing:** Danielle S. LeSassier, Kathleen Q. Schulte, Tara E. Manley, Alan R. Smith, Megan L. Powals, Nicolette C. Albright, Benjamin C. Ludolph, Katharina L. Weber, Myles W. Gardner, F. Curtis Hewitt.

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
