## [Decision Letter · Decision Letter 0]

9 Jul 2019

PONE-D-19-16966

Artificial fingerprints for cross-comparison of forensic DNA and protein recovery methods

PLOS ONE

Dear Dr. Hewitt,

Thank you for submitting your manuscript to PLOS ONE. After careful consideration, we feel that it has merit but does not fully meet PLOS ONE’s publication criteria as it currently stands. Therefore, we invite you to submit a revised version of the manuscript that addresses the points raised by the two reviewers during the review process (as shown below).

We would appreciate receiving your revised manuscript by Aug 23 2019 11:59PM. To enhance the reproducibility of your results, we recommend that if applicable you deposit your laboratory protocols in protocols.io, where a protocol can be assigned its own identifier (DOI) such that it can be cited independently in the future. For instructions see: http://journals.plos.org/plosone/s/submission-guidelines#loc-laboratory-protocols

We look forward to receiving your revised manuscript.

Kind regards,

Andy T. Y. Lau, Ph.D.

Academic Editor

PLOS ONE

**Journal Requirements:**

2. We kindly ask that you address the difference in ethics committee and authors affiliations. In your Ethics Statement you name the experimental animal ethics committee at the University of North Texas Health Science Center as the institutional review board (IRB) that approved your work. However, none of the authors are affiliated with University of North Texas. We would generally expect at least one author to be affiliated with the institution to which the IRB belongs. Can you please explain why this is not the case?

3. Additionally, as your study include human participants, we would be grateful if for reproducibility reasons you could amend your Methods section to include details of how and when human subjects' samples were collected and the date(s) when you conducted this study. Please also include basic demographic details such age and gender for reasons of reproducibility. Thank you for considering our requests.

4.  We note that one or more of the authors are employed by a commercial company: "Signature Science."

**Comments to the Author**

1. Is the manuscript technically sound, and do the data support the conclusions?

Reviewer #1: Partly

Reviewer #2: Partly

2. Has the statistical analysis been performed appropriately and rigorously? 

Reviewer #1: No

Reviewer #2: No

3. Have the authors made all data underlying the findings in their manuscript fully available?

Reviewer #1: Yes

Reviewer #2: Yes

4. Is the manuscript presented in an intelligible fashion and written in standard English?

Reviewer #1: Yes

Reviewer #2: Yes

5. Review Comments to the Author

Reviewer #1: In the current manuscript, the authors described a method to generate customizable artificial fingerprints that can be used for benchmarking fingerprint collection and extraction processes. Overall, the manuscript is well-written and easy to follow.

By comparing the content and quality of DNA/protein and proteomic profiles in human and artificial fingerprints, the authors concluded that the artificial fingerprints they produced could benefit the laboratory work of forensic casework. However, I have several concerns and questions regarding this work:

1. No statistical analysis mentioned.

2. How many samples were tested for each of the experiments? How many times were the experiments repeated?

3. According to the data shown in Figure 3B, it seems that there are limited overlapping proteins in human and artificial fingerprints, in both metal and glass surface. If so, how could you claim that the proteomic profiles of human and artificial fingerprints are similar?

4. Why only tested on glass and metal surfaces?

5. What “metal” did you test on?

6. The usability of the artificial fingerprints in forensic field is still unclear – the authors should test (or at least discuss) the potential downsides of these artificial fingerprints. E.g., storage? Degradation rate of DNA/protein?

7. The duplicate loadings in Figure 2C did not seem to be equal.

Reviewer #2: It is an interesting work and may be useful to better assess recovery techniques for touch depositions. However, I have some questions and considerations toward these work:

1. How many donors or volunteers in your study and what is the inclusion criteria?

2. There is no statistical analysis in the study.

3. If your artificial fingerprint products be used commercially in the future, what measures will you take to control the products' quality? Eg. How much %-matched for the gene/protein profile will be consiered of “good-quality”?

6. PLOS authors have the option to publish the peer review history of their article (what does this mean?). If published, this will include your full peer review and any attached files.

Reviewer #1: No

Reviewer #2: No

---

## [Author Response · Author response to Decision Letter 0]

22 Jul 2019

We appreciate the thoughtful comments from the editor and the reviewers and feel this has strengthened our manuscript. We are resubmitting a revised version of our research article “Artificial fingerprints for cross-comparison of forensic DNA and protein recovery methods”. Based on reviewers’ comments, we have included more detailed statistical and sample information, addressed limitations regarding the artificial fingerprints in the discussion, and clarified several data result sections. Specific feedback is included below (as well as in the cover letter and response to reviewers document). 

Reviewer #1: In the current manuscript, the authors described a method to generate customizable artificial fingerprints that can be used for benchmarking fingerprint collection and extraction processes. Overall, the manuscript is well-written and easy to follow.

By comparing the content and quality of DNA/protein and proteomic profiles in human and artificial fingerprints, the authors concluded that the artificial fingerprints they produced could benefit the laboratory work of forensic casework. However, I have several concerns and questions regarding this work:

1. No statistical analysis mentioned.

We have updated our plots and text to include additional statistical information, including sample number, individual replicates, and the mean ± standard deviation, and described this in the text where relevant (see figure legends for Figures 1 and 2).

2. How many samples were tested for each of the experiments? How many times were the experiments repeated?

We have added the requested information into the figure legends (see figure legends for Figures 1 and 2).

3. According to the data shown in Figure 3B, it seems that there are limited overlapping proteins in human and artificial fingerprints, in both metal and glass surface. If so, how could you claim that the proteomic profiles of human and artificial fingerprints are similar?

We respectfully disagree with the reviewer and believe we have provided sufficient evidence supporting concordance between the proteomic profiles of artificial prints and latent prints. Table 1 demonstrates the high overlap in the top 10 most abundant proteins found in the two sample types and the Figure 3A heatmap shows that the top 50 proteins have reasonably equivalent sequence coverage and number of peptides between artificial and latent fingerprints collected from both surface types. This supports that the overlapping 61 proteins are likely to represent many of the key touch proteins present and unique hits may represent lower abundance proteins. We have added a citation for the explanation regarding the unique protein IDs (Line 336, reference [21] Liu et al. (2004) Anal Chem) and added the following text.

Lines 336-338: Overall, the similarities in presence, sequence coverage, and number of peptides in the most abundant proteins between latent and artificial fingerprint samples supports concordance in the proteomic profiles.

4. Why only tested on glass and metal surfaces?

These are forensically-relevant, non-porous materials and have noted that in the text.

Lines 225-227: Latent, loaded, and artificial prints (containing either 5 or 10 ng DNA) were deposited onto glass or chrome metal surfaces, two non-porous surfaces of forensic value, collected, and extracted for DNA quantitation.

5. What “metal” did you test on?

All samples were collected from chrome metal plates. We have updated the text to clarify this.

6. The usability of the artificial fingerprints in forensic field is still unclear – the authors should test (or at least discuss) the potential downsides of these artificial fingerprints. E.g., storage? Degradation rate of DNA/protein?

We have added additional text to the discussion to address this limitation.

Lines 368-375: The current study does not explore the stability of artificial fingerprints, in-solution or as a deposition, opting to make each sample fresh prior to collection. As such, it is currently unclear what potential impacts long-term storage may have onto the integrity of the DNA or protein. Being able to stably store these artificial fingerprints without DNA or protein degradation is an important factor for forensic application. Proteins are inherently more environmentally robust biomolecules than DNA [23,24], but how the artificial sebaceous/perspiration mixture affects either component under long-term storage conditions has not yet been defined. Stability studies of both in-solution and deposited artificial fingerprints are of future interest.

23. Lindahl T. Instability and decay of the primary structure of DNA. Nature. 1993 Apr;362(6422):709–15. 

24. Wadsworth C, Buckley M. Proteome degradation in fossils: investigating the longevity of protein survival in ancient bone: Proteome degradation in fossils. Rapid Commun Mass Spectrom. 2014 Mar 30;28(6):605–15. 

7. The duplicate loadings in Figure 2C did not seem to be equal.

While the homogenization process greatly reduces ESM particulate size variability, it does not completely remove all larger pieces, which can cause some variability during deposition. We have added additional text to clarify this point.

Lines 260-262: While this step greatly improved pipetting the artificial fingerprints, it did not completely alleviate issues associated with larger ESM particulates causing clogging or uneven deposition.

Lines 284-286: While some variability was observed between replicates, possibly due to larger particulate deposition not completely removed during the homogenization step, the 12.5 µg deposition appeared to most closely match the protein levels in a latent fingerprint.

 

Reviewer #2: It is an interesting work and may be useful to better assess recovery techniques for touch depositions. However, I have some questions and considerations toward these work:

1. How many donors or volunteers in your study and what is the inclusion criteria?

We have added additional detail to the Methods.

Lines 82 – 84: Twenty-five adult (over 18 years old) male and female donors of northern European ancestry provided ESM samples for proteomic analysis by Ped Egg collection, as described above, and stored at -80 oC. 

2. There is no statistical analysis in the study.

We have updated our plots and text to include additional statistical information, stating sample number, representing individual replicates, and showing the mean ± standard deviation. This is described in the text where relevant (see figure legends for Figures 1 and 2).

3. If your artificial fingerprint products be used commercially in the future, what measures will you take to control the products' quality? Eg. How much %-matched for the gene/protein profile will be consiered of “good-quality”?

Commercialization is not an avenue we had considered and is not discussed in the manuscript. The reviewer raises valid points about quality control hurdles should we choose to pursue commercialization, but we feel it is outside the scope of the current manuscript.

 

Academic Editor:

Author responses are in red text.

2. We kindly ask that you address the difference in ethics committee and authors affiliations. In your Ethics Statement you name the experimental animal ethics committee at the University of North Texas Health Science Center as the institutional review board (IRB) that approved your work. However, none of the authors are affiliated with University of North Texas. We would generally expect at least one author to be affiliated with the institution to which the IRB belongs. Can you please explain why this is not the case?

We have included Dr. August Woerner from University of North Texas as an author.

3. Additionally, as your study include human participants, we would be grateful if for reproducibility reasons you could amend your Methods section to include details of how and when human subjects' samples were collected and the date(s) when you conducted this study. Please also include basic demographic details such age and gender for reasons of reproducibility. Thank you for considering our requests.

We have added additional detail to the Methods.

Lines 82 – 84: Twenty-five adult (over 18 years old) male and female donors of northern European ancestry provided ESM samples for proteomic analysis by Ped Egg collection, as described above, and stored at -80 oC. 

4. We note that one or more of the authors are employed by a commercial company: "Signature Science."

We have added the requested funding language to the Funding Statement:

Lines 408-412: This research was supported in part by internal funding from Signature Science, LLC. Signature Science, LLC provided support in the form of salaries for authors DSL, KQS, TEM, ARS, MLP, NCA, BCL, KLW, MWG, and FCH but did not have any additional role in the study design, data collection and analysis, decision to publish, or preparation of the manuscript. The specific roles of these authors are articulated in the ‘author contributions’ section.

We have updated the Competing Interests and added the requested funding language:

Lines 415-419: The authors DSL, KQS, TEM, ARS, MLP, NCA, BCL, KLW, MWG, and FCH are employed by Signature Science, LLC. This does not alter our adherence to PLOS ONE policies on sharing data and materials. The authors declare no other relevant affiliations or financial involvement with a financial interest in or financial conflict with the subject matter or materials discussed in the manuscript apart from those disclosed.

---

## [Decision Letter · Decision Letter 1]

17 Sep 2019

Artificial fingerprints for cross-comparison of forensic DNA and protein recovery methods

PONE-D-19-16966R1

Dear Dr. Hewitt,

We are pleased to inform you that your manuscript has been judged scientifically suitable for publication and will be formally accepted for publication once it complies with all outstanding technical requirements.

With kind regards,

Andy T. Y. Lau, Ph.D.

Academic Editor

PLOS ONE

Additional Editor Comments (optional):

Reviewers' comments:

Reviewer's Responses to Questions

**Comments to the Author**

1. If the authors have adequately addressed your comments raised in a previous round of review and you feel that this manuscript is now acceptable for publication, you may indicate that here to bypass the “Comments to the Author” section, enter your conflict of interest statement in the “Confidential to Editor” section, and submit your "Accept" recommendation.

Reviewer #1: All comments have been addressed

Reviewer #2: All comments have been addressed

2. Is the manuscript technically sound, and do the data support the conclusions?

Reviewer #1: Yes

Reviewer #2: Yes

3. Has the statistical analysis been performed appropriately and rigorously? 

Reviewer #1: Yes

Reviewer #2: Yes

4. Have the authors made all data underlying the findings in their manuscript fully available?

Reviewer #1: Yes

Reviewer #2: Yes

5. Is the manuscript presented in an intelligible fashion and written in standard English?

Reviewer #1: Yes

Reviewer #2: Yes

6. Review Comments to the Author

Reviewer #1: All the comments and concerns from the both reviewers have been appropriately addressed by the authors. I have no further question.

Reviewer #2: (No Response)

7. PLOS authors have the option to publish the peer review history of their article (what does this mean?). If published, this will include your full peer review and any attached files.

Reviewer #1: No

Reviewer #2: No